

# Influence of isoflurane on the diastolic pressure-flow relationship and critical occlusion pressure during arterial CABG surgery: a randomized controlled trial

José Hinz[1], Ashham Mansur[1], Gerd G. Hanekop[2], Andreas Weyland[3], Aron F. Popov[4], Jan D. Schmitto[5], Frank F. G. Grüne[6], Martin Bauer[7] and Stephan Kazmaier[2]

[1] Department of Anesthesiology, University Medical Center Goettingen, Germany
[2] Department of Anesthesiology, University Medical Center Goettingen, Goettingen, Germany
[3] Department of Anesthesiology and Intensive Care Medicine, University of Oldenburg, Oldenburg, Germany
[4] Department of Cardiothoracic Surgery, Transplantation & Mechanical Support, Royal Brompton & Harefield Hospital, London, United Kingdom
[5] Department of Cardiothoracic, Transplant and Vascular Surgery, Hannover Medical School, Hannover, Germany
[6] Department of Anesthesiology, Erasmus University/Rotterdam, Rotterdam, Netherlands
[7] Department of Anesthesiology, University Medical Center Goettingen, Göttingen, Germany

Corresponding author
Ashham Mansur, ashham.mansur@med.uni-goettingen.de

## ABSTRACT

The effects of isoflurane on the determinants of blood flow during Coronary Artery Bypass Graft (CABG) surgery are not completely understood. This study characterized the influence of isoflurane on the diastolic Pressure-Flow (P-F) relationship and Critical Occlusion Pressure (COP) during CABG surgery. Twenty patients undergoing CABG surgery were studied. Patients were assigned to an isoflurane or control group. Hemodynamic and flow measurements during CABG surgery were performed twice (15 minutes after the discontinuation of extracorporeal circulation (T15) and again 15 minutes later (T30)). The zero flow pressure intercept (a measure of COP) was extrapolated from a linear regression analysis of the instantaneous diastolic P-F relationship. In the isoflurane group, the application of isoflurane significantly increased the slope of the diastolic P-F relationship by 215% indicating a mean reduction of Coronary Vascular Resistance (CVR) by 46%. Simultaneously, the Mean Diastolic Aortic Pressure (MDAP) decreased by 19% mainly due to a decrease in the systemic vascular resistance index by 21%. The COP, cardiac index, heart rate, Left Ventricular End-Diastolic Pressure (LVEDP) and Coronary Sinus Pressure (CSP) did not change significantly. In the control group, the parameters remained unchanged. In both groups, COP significantly exceeded the CSP and LVEDP at both time points. We conclude that short-term application of isoflurane at a sedative concentration markedly increases the slope of the instantaneous diastolic P-F relationship during CABG surgery implying a distinct decrease with CVR in patients undergoing CABG surgery.

## INTRODUCTION

In a theoretical approach to the Pressure-Flow (P-F) relationship in arterioles, these calculations are a simplification of the actual variable tissue characteristics in the vascular bed of an organ (*Hoffman & Spaan, 1990*). We demonstrated in an earlier study on the diastolic coronary P-F relationship that the effective downstream pressure is not determined by Coronary Sinus Pressure (CSP) or Left Ventricular End-Diastolic Pressure (LVEDP) but by the Critical Occlusion Pressure (COP) of the coronary vasculature, which was considerably higher than both parameters (*Kazmaier et al., 2006*). The zero flow pressure intercept as a measure of COP was extrapolated from the linear regression analysis of the instantaneous diastolic P-F relationship. However, the effects of isoflurane on the determinants of blood flow during Coronary Artery Bypass Graft (CABG) surgery are not completely understood. Earlier investigations yielded discrepant findings on the risk of myocardial ischemia due to the vasoactive potency of isoflurane. Some investigations found that the risk of myocardial ischemic events due to coronary flow misdistribution is increased when anesthesia is maintained with isoflurane (*Buffington et al., 1987*; *Diana et al., 1993*; *Inoue et al., 1990*; *Khambatta et al., 1988*; *Priebe & Foex, 1987*). In contrast, results from other investigators demonstrated that the risk of perioperative myocardial ischemia was not increased during isoflurane anesthesia compared with other volatile anesthetics or total intravenous anesthetic regimens (*Leung et al., 1991*; *Pulley et al., 1991*). Furthermore, in some studies the vasoactive potency of isoflurane positively affected the regional distribution of coronary blood flow (*Hartman et al., 1990*; *Kim et al., 1994*). However, regarding volatile anesthetics, maximal increases in global coronary blood flow were obtained during isoflurane anesthesia (*Crystal et al., 2000*). According to the fact that volatile anesthetics (*Landoni et al., 2013*) and in particular isoflurane are beneficial for myocardial ischemia and has been shown to improve survival in cardiac surgery (*Bignami et al., 2013*; *Chiari et al., 2005*; *Ge et al., 2010*; *Lang et al., 2013*), this study aimed at investigating the impact of isoflurane on the diastolic P-F relationship and COP during CABG surgery.

## PATIENTS AND METHODS

### Patients

This study was approved by the University of Goettingen ethics committee in Goettingen, Germany (12/4/04) and conformed to the ethical principles of the Declaration of Helsinki. Written informed consent was obtained from all patients. Twenty patients (17 males and 3 females) with angiographically verified Coronary Artery Disease (CAD) were studied following elective CABG surgery. Biometric and intraoperative data are presented in Table 1. Patients with concomitant valvular heart disease or a lack of sinus rhythm at the start of the measurement period were excluded from this study. Antiarrhythmic and

**Table 1 Biometric and intraoperative data.** Results are presented as medians and ranges, if required.

| | Control (n = 10) | Isoflurane (n = 10) | p-value |
|---|---|---|---|
| Age (years) | 70 (58–79) | 71 (51–81) | 0.44 |
| Weight (kg) | 80 (60–92) | 75 (60–103) | 0.68 |
| Height (cm) | 168 (153–82) | 168 (156–180) | 0.68 |
| Gender (male/female) | 8/2 | 9/1 | 0.53 |
| Body surface area (m$^2$) | 1.89 (1.57–2.12) | 1.89 (1.59–2.09) | 0.97 |
| Coronary artery disease | 3 vessel disease, n = 7<br>2 vessel disease, n = 3 | 3 vessel disease, n = 6<br>2 vessel disease, n = 3<br>1 vessel disease, n = 1 | |
| Grafts (n) | 4 (3–5) | 4 (2–5) | |
| Aortic clamping time (minutes) | 78 (61–92) | 85 (45–112) | 0.68 |
| Reperfusion time (minutes) | 44 (32–65) | 40 (26–66) | 0.48 |

antihypertensive medications (except Angiotensin-Converting Enzyme (ACE) inhibitors) were continued until the day of surgery. The pre-anesthetic medication consisted of 1.0 mg of flunitrazepam per os on the evening prior to surgery and 60 minutes before the induction of general anesthesia. Patients were pre-operatively randomly assigned to receive either total intravenous anesthesia during the complete study period (control group) or an additional 0.4% volume of isoflurane (1.0 Minimum Anesthetic Concentration (MAC)$_{sedative}$, isoflurane group) immediately after the baseline measurements.

## Methods

Before the induction of anesthesia, electrocardiogram leads were placed, and a 15-gauge catheter (PV2015L20; Pulsion GmbH, Munich, Germany) was placed in the femoral artery to measure the Mean Arterial Pressure (MAP). Cardiac output measurements were performed with a transpulmonal thermodilution technique, pulse contour analysis (PiCCO, Pulsion GmbH, Munich, Germany) and blood sampling. Intravenous induction of anesthesia was performed with 2.0 $\mu$g kg$^{-1}$ sufentanil and 0.1 mg kg$^{-1}$ pancuronium bromide to facilitate endotracheal intubation. Subsequently, a central venous catheter (8.5 Fr., 4-lumen; ARROW GmbH, Erding, Germany) was inserted via the right internal jugular vein to measure Central Venous Pressure (CVP) and for drug and fluid administration.

Maintenance of total intravenous anesthesia was performed by continuous infusion of sufentanil (3.0 $\mu$g kg$^{-1}$ h$^{-1}$) and additive boluses of midazolam if necessary. The patients' lungs were ventilated by intermittent positive pressure ventilation (Cicero; Draeger GmbH, Lübeck, Germany). The respiratory rate and minute volume were adjusted to achieve normocapnia. During the measurement period, the fraction of inspired oxygen (F$_i$O$_2$) was 0.5 to avoid hypoxemic events.

Extracorporeal circulation with standard techniques included two-stage venous cannulation (Medtronic MC2TM, 91246C; 34/46Fr.; Medtronic Inc., Minneapolis, MN, USA), central aortic cannulation (Aortic Arch Cannula-Straight/Wire Inlay, 6.5 mm,
A232–65; Stöckert Instrumente GmbH, Munich, Germany) and membrane oxygenation (Hilite® 7000; Medizintechnik AG, Stolberg, Germany). Surgery was performed during aortic cross-clamping and cardioplegic arrest with combined anterograde and retrograde cold blood cardioplegia (Dr. Franz Köhler Chemie GmbH, Alsbach-Hähnlein, Germany).

## Measurements

The Coronary Vascular Resistance (CVR) was calculated as the ratio of Coronary Perfusion Pressure (CPP) and coronary blood flow. CPP was calculated as the difference between the upstream pressure and downstream pressure. The coronary upstream pressure was expressed as the mean diastolic aortic pressure. CSP and LVEDP are generally used as equivalents of coronary downstream pressure (reflecting the specific characteristics of the coronary anatomy).

Baseline measurements were performed 15 minutes ($T_{15}$) after the discontinuation of extracorporeal circulation under steady state conditions. Fifteen minutes after the end of the first measurement period ($T_{30}$), a second measurement period was performed in which patients in the isoflurane group received 0.4% volume of isoflurane for 15 minutes. In the control group, the anesthetic regimen did not change between the two measurement periods. All of the measurements were performed prior to the reversal of heparin. At the beginning of each measurement period, cardiac output was assessed by the transpulmonal thermodilution method as the mean of three injections of 15 ml of ice-cooled isotonic saline solution randomly distributed over the respiratory cycle. During the short period (5 to 10 seconds) of coronary blood flow measurements, the patient was disconnected from the ventilator to avoid intrapleural pressure changes and volume shifts that might influence the P-F relationships. In this study, to avoid capacitance effects of the arterial vessel by opening phenomena, the analysis of the P-F relationship only included data from the highest diastolic flow rate in the arterial bypass graft until the end of diastole.

Flow measurements were performed using ultrasound and calculations based on the transit time principle (Cardiomed 4008, Quick-Fit probes (size 2.0–3.0 mm); Medistim, Oslo, Norway). The flow in the left internal mammary artery and pressure measurements were recorded simultaneously over a period of 5 seconds using analogue-digital converting devices with a sampling frequency of 500 Hz.

The measurements of aortic pressure and CSP were recorded via the cannulae for the extracorporeal circuit and retrograde blood cardioplegia, respectively (coronary sinus cannula: retrograde cardioplegia cannula RSH-M014S, 14 Fr; Chase Medical, Richardson, Texas, USA). LVEDP values were obtained using a left atrial catheter (Jostra KLAP1751 pressure monitoring catheter, 5.0 Fr; Jostra AG, Hirrlingen, Germany) introduced via the upper right pulmonary vein and positioned trans-mitrally into the left ventricle during the study period.

Arterial and coronary sinus blood samples were obtained immediately after each period of coronary blood flow measurement for the determination of pH ($pH_{art}$, $pH_{cs}$), acid base status, arterial and coronary sinus blood gas tensions ($P_aO_2$, $P_{cs}O_2$, $P_aCO_2$, $P_{cs}CO_2$), oxygen saturation ($S_aO_2$, $S_{cs}O_2$), hemoglobin (Hb) and lactate concentrations
(ABL 700; Radiometer Medical A/S, Denmark). In the isoflurane group, additional coronary sinus blood samples were obtained after the second measurement period for the determination of isoflurane plasma concentrations.

## Calculations

The Cardiac Index (CI), Stroke Volume Index (SVI) and Systemic Vascular Resistance Index (SVRI) were calculated according to standard formulae. The critical occlusion pressure was calculated by extrapolation of the linear regression analysis of the diastolic part of the aortic P-F loop to the zero flow pressure intercept (2). The mean LVEDP was assessed by analysis of the left ventricular pressure curve for every beat during the study period. The CSP was calculated as the mean CSP of the entire period of flow measurements.

## Statistics

Statistical procedures were performed using the Statistical Package for the Social Sciences (SPSS) (SPSS 17.0 for MAC; SPSS Inc., Chicago, Illinois, USA). Results are expressed as medians and ranges. Two continuous variables were compared using Mann-Whitney test. Linear regression analysis was performed using flow as the dependent variable and diastolic aortic pressure as the independent variable. A p-value less than 0.05 was considered to be significant.

# RESULTS

The biometric and perioperative data of the studied patients were not different between both groups (Table 1). The data from five consecutive heart beats were analyzed for each patient; none of the patients had to be excluded due to missing data or artifacts in hemodynamics and flow measurements.

## Systemic hemodynamics

In both groups, GEDI and SVV (parameters of volume status), as well as LVEDP, CFI, and EVLWI, did not change during the entire measurement period and were not different between groups. In the control group, MAP, CI, HR and SVRI did not change during the entire study period. In contrast to measurements in the control group, the application of 1.0 $MAC_{sedative}$ isoflurane decreased MAP from 69 to 57 mmHg (17%) and SVRI from 1730 to 1364 dyn sec $cm^{-5}$ $m^2$ (21%) but had no effect on HR and CI. The systemic hemodynamic data are summarized in Tables 2 and 3.

The diastolic flow and COP during CABG surgery did not differ between measurement periods or groups. In the control group, COP exceeded LVEDP by 244% and 280%, respectively. In the control group, COP exceeded CSP by 279% and 323%, respectively. In the isoflurane group, similar results were found with 318%, 262%, 350% and 340%, respectively.

The application of isoflurane decreased the MDAP from 62 mmHg to 50 mmHg, whereas MDAP did not change in the control group.

Consequently, CVR changes were only observed in the isoflurane group (if calculated using COP and LVEDP but not CSP). Considerable differences in CVR were observed depending on the downstream pressure (COP, LVEDP or CSP) used in the formula.

**Table 2 Hemodynamic data.** Results are presented as medians and ranges.

| | Control | | | Isoflurane | | |
|---|---|---|---|---|---|---|
| | $T_{15}$ | $T_{30}$ | Control T15 vs $T_{30}$ (p-value) | $T_{15}$ | $T_{30}$ | Isoflurane T15 vs $T_{30}$ (p-value) |
| CI (l min$^{-1}$ m$^{-2}$) | 2.9 (2.3–3.6) | 2.7 (2.1–3.2) | 0.11 | 2.7 (1.6–5.7) | 2.6 (1.9–5.8) | 0.28 |
| HR (min$^{-1}$) | 89 (53–106) | 90 (56–106) | 0.65 | 99 (76–112) | 88 (74–100) | 0.15 |
| MAP (mmHg) | 65 (47–80) | 66 (56–84) | 0.06 | 69 (59–85) | 57 (48–59) | 0.005 |
| MDAP (mmHg) | 57 (45–64) | 60 (43–71) | 0.06 | 62 (53–79) | 50 (45–57) | 0.005 |
| DDT (ms) | 447 (257–1215) | 489 (366–1070) | 0.60 | 721 (240–919) | 567 (339–937) | 0.33 |
| CVP (mmHg) | 9 (7–16) | 9 (7–16) | *1.0* | 10 (7–12) | 10 (7–13) | 0.82 |
| CSP (mmHg) | 14 (8–18) | 13 (7–18) | 0.39 | 10 (5–20) | 10 (6–19) | 0.44 |
| SVRI (dyn sec cm$^{-5}$ m$^2$) | 1555 (809–2454) | 1716 (1355–2193) | 0.31 | 1730 (850–2969) | 1364 (699–1965) | 0.04 |
| GEDI (ml m$^{-2}$) | 642 (541–895) | 652 (594–883) | 0.78 | 642 (457–834) | 617 (479–802) | 0.92 |
| EVLWI (ml m$^{-2}$) | 6.8 (5.7–10.0) | 6.3 (5.0–8.7) | 0.62 | 7.4 (4.0–10.7) | 7.0 (4.3–10.0) | 0.16 |
| LVEDP (mmHg) | 16 (8–33) | 15 (8–30) | 0.09 | 11 (8–37) | 13 (9–22) | 0.80 |
| SVV (%) | 11 (6–17) | 12 (5–18) | 0.64 | 12 (5–19) | 14 (8–16) | 0.43 |
| dp/dt$_{max}$ (mmHg s$^{-1}$) | 766 (507–1063) | 763 (500–887) | 0.65 | 748 (510–1350) | 651 (340–1143) | 0.02 |
| CFI (min$^{-1}$) | 4.2 (3.5–5.0) | 3.7 (2.8–5.0) | 0.05 | 4.6 (2.5–8.1) | 3.9 (3.1–7.4) | 0.16 |
| COP (mmHg) | 39 (23–48) | 42 (28–44) | 0.25 | 35 (26–59) | 34 (27–42) | 0.11 |
| FLOW (ml/min) | 26 (10–38) | 22 (9–53) | 0.72 | 29 (7–68) | 23 (11–50) | 0.24 |
| Slope (B1) | 1.36 (0.48–2.71) | 1.29 (0.46–2.10) | 0.49 | 0.94 (0.50–4.82) | 2.03 (0.56–4.41) | 0.02 |
| CVR from Slope 1/B1 | 0.74 (0.37–2.10) | 0.78 (0.48–2.17) | 0.49 | 1.06 (0.21–2.02) | 0.51 (0.23–1.79) | 0.007 |
| CVR from CSP | 1.73 (1.21–3.92) | 1.83 (0.94–5.51) | 0.69 | 1.80 (0.78–8.83) | 1.35 (0.81–3.35) | 0.30 |
| CVR from LVEDP | 1.66 (0.81–4.05) | 1.86 (0.61–5.64) | 0.49 | 1.68 (0.79–5.38) | 1.77 (0.73–3.13) | 0.06 |
| CVR from COP | 0.74 (0.37–2.07) | 0.78 (0.46–2.13) | 0.44 | 1.06 (0.21–2.02) | 0.51 (0.23–1.79) | 0.006 |

With CSP as a measure of downstream pressure, CVR did not change significantly in either group. In the control group, the CVR values were 1.73 and 1.83 mmHg ml$^{-1}$ minute, while those in the isoflurane group were 1.8 and 1.35 mmHg ml$^{-1}$ minute. Similar findings were observed for CVR calculated with LVEDP in both groups.

In contrast to these findings, using COP in the formula indicated that CVR was decreased after the application of isoflurane from 1.06 to 0.51 mmHg ml$^{-1}$ minute. In the control group, CVR was calculated using COP, and no differences were observed (0.74 and 0.78 mmHg ml$^{-1}$ minute). As expected, using COP for the calculation of CVR resulted in significantly lower values for CVR than using LVEDP or CSP. The CVR values were identical when calculated according to the slope of the instantaneous diastolic P-F relationship. The diastolic flow, COP and CVR calculations are summarized in Tables 2 and 3.

Arterial blood gas analyses and acid base status also did not differ between groups and remained unchanged during the measurement period. For the coronary sinus blood gas samples, $S_{cs}O_2$ $p_{cs}O_2$, $p_{cs}CO_2$, $pH_{cs}$, $BE_{cs}$ and $SBIC_{cs}$ and lactate did not differ between the measurement periods in the control group. In contrast to these findings, the application of isoflurane led to a significant increase in $S_{cs}O_2$ and $p_{cs}O_2$ by 22% and 22%, respectively, indicating a reduced myocardial oxygen extraction. This decrease in myocardial oxygen

**Table 3  Hemodynamic data.** Isoflurane vs control.

| | Control vs Isoflurane T$_{15}$ (p-value) | Control vs Isoflurane T$_{30}$ (p-value) |
|---|---|---|
| CI (l min$^{-1}$ m$^{-2}$) | 0.96 | 0.90 |
| HR (min$^{-1}$) | 0.90 | 0.85 |
| MAP (mmHg) | 0.06 | 0.009 |
| MDAP (mmHg) | 0.06 | 0.01 |
| DDT (ms) | 0.56 | 0.90 |
| CVP (mmHg) | 0.96 | 0.83 |
| CSP (mmHg) | 0.16 | 0.11 |
| SVRI (dyn sec cm$^{-5}$ m$^2$) | 0.51 | 0.05 |
| GEDI (ml m$^{-2}$) | 0.69 | 0.27 |
| EVLWI (ml m$^{-2}$) | 0.31 | 0.41 |
| LVEDP (mmHg) | 0.34 | 0.47 |
| SVV (%) | 0.96 | 0.57 |
| dp/dtmax (mmHg s$^{-1}$) | 0.96 | 0.57 |
| CFI (min$^{-1}$) | 0.56 | 0.76 |
| COP (mmHg) | 0.87 | 0.11 |
| FLOW (ml/min) | 0.38 | 0.79 |
| Slope (B1) | 0.79 | 0.05 |
| CVR from Slope 1/B1 | 0.87 | 0.14 |
| CVR from CSP | 0.83 | 0.28 |
| CVR from LVEDP | 0.85 | 0.39 |
| CVR from COP | 0.73 | 0.10 |

extraction was not associated with a significant change in MLE. The arterial and coronary sinus blood gas analysis results are presented in Tables 4 and 5.

## DISCUSSION

This study investigated the influence of short-term isoflurane administration on the instantaneous diastolic P-F relationship for the calculation of CVR and COP in patients undergoing elective CABG surgery. CVR was calculated with different techniques. First, CVR was calculated with conventional formulas using MDAP as the upstream pressure and LVEDP or CSP as the downstream pressures. Second, CVR was calculated from the instantaneous diastolic P-F relationship using either the slope of the linear diastolic portion or the linear extrapolation of this slope to zero flow (with COP for the calculation of downstream pressure). COP was about two to three times higher than the generally used downstream pressures (CSP or LVEDP) and was not influenced by isoflurane. This result may be explained by a waterfall phenomenon in the coronary circulation (*Kazmaier et al., 2006*; *Maas et al., 2012*). In addition, CVR decreased following the application of isoflurane only when the instantaneous P-F relationship was used in the calculation. The CVR decreases derived from the P-F relationship showed excellent concordance. The finding of reduced CVR was supported by decreased myocardial oxygen extraction in the isoflurane group.

**Table 4 Arterial and coronary sinus blood gas analyses.** Results are presented as medians and ranges.

| | Control | | | Isoflurane | | |
|---|---|---|---|---|---|---|
| | $T_{15}$ | $T_{30}$ | Control $T_{15}$ vs $T_{30}$ | $T_{15}$ | $T_{30}$ | Isoflurane $T_{15}$ vs $T_{30}$ |
| $Hb_{art}$ (g/dl) | 9.1 (7.5–10.0) | 9.0 (7.8–10.9) | 0.51 | 9.2 (8.1–10.3) | 9.2 (7.6–11.6) | 0.91 |
| $S_aO_2$ (%) | 99 (94.1–99.6) | 99 (93.7–99.9) | 1.0 | 99 (98.7–99.7) | 99 (97.2–99.7) | 0.31 |
| $P_aO_2$ (mmHg) | 163 (71–298) | 153 (75–364) | 0.43 | 200 (124–293) | 213 (141–277) | 0.28 |
| $P_aCO_2$ (mmHg) | 46 (38–54) | 48 (39.7–51.7) | 0.08 | 43 (35–55) | 44 (34–50) | 0.82 |
| $pH_{art}$ | 7.35 (7.24–7.41) | 7.35 (7.22–7.38) | 0.08 | 7.35 (7.24–7.44) | 7.34 (7.28–7.45) | 1.0 |
| $BE_{art}$ (mmol/l) | −0.4 (−7.6–1.3) | −1.2 (−7.2–1.1) | 0.55 | −1.4 (−4.4–1.2) | −2.4 (−3.6–0.1) | 0.31 |
| $SBIC_{art}$ (mmol/l) | 24.1 (18.2–25.5) | 23.5 (18.5–25.5) | 0.58 | 22.7 (20.7–24.2) | 22.4 (21.4–24.6) | 0.43 |
| $S_{cs}O_2$ (%) | 49 (23.5–82.9) | 46 (39.3–74.9) | 0.49 | 46 (29.7–74.5) | 56 (33.1–80.3) | 0.05 |
| $P_{cs}O_2$ (mmHg) | 26 (19–51) | 27 (24–43) | 0.70 | 27 (19–44) | 33 (22–49) | 0.006 |
| $P_{cs}CO_2$ (mmHg) | 56 (38–64) | 58 (48–62) | 0.06 | 54 (38–66) | 55 (40–64) | 1.0 |
| $pH_{cs}$ | 7.30 (7.14–7.35) | 7.30 (7.17–7.32) | 0.13 | 7.30 (7.19–7.42) | 7.28 (7.23–7.40) | 0.30 |
| $BE_{cs}$ (mmol/l) | 0.3 (−7.9–2.0) | −0.1 (−6.8–1.9) | 0.91 | −1.4 (−3.9–0.7) | −2.2 (−3.4–0.2) | 0.02 |
| $SBIC_{cs}$ (mmol/l) | 23.9 (17.0–25.3) | 23.7 (18.1–25.2) | 0.84 | 22.6 (20.8–24.3) | 21.9 (21.3–24.2) | 0.03 |
| $Lactate_{cs}$ (mmol/l) | 1.0 (0.6–5.8) | 1.0 (0.5–5.3) | 0.16 | 1.8 (1.0–3.0) | 2.0 (0.8–2.7) | 0.39 |
| $Isoflurane_{cs}$ (μg/dl) | 0 | 0 | | 0 | 1.4 (0.8–2.3) | |

**Table 5 Arterial and coronary sinus blood gas analyses.** Isoflurane vs Control.

| | Control vs Isoflurane $T_{15}$ (p-value) | Control vs Isoflurane $T_{30}$ (p-value) |
|---|---|---|
| $Hb_{art}$ (g/dl) | 0.43 | 0.57 |
| $S_aO_2$ (%) | 0.17 | 0.32 |
| $P_aO_2$ (mmHg) | 0.32 | 0.08 |
| $P_aCO_2$ (mmHg) | 0.39 | 0.12 |
| $pH_{art}$ | 0.35 | 0.68 |
| $BE_{art}$ (mmol/l) | 0.22 | 0.25 |
| $SBIC_{art}$ (mmol/l) | 0.19 | 0.25 |
| $S_{cs}O_2$ (%) | 0.85 | 0.17 |
| $P_{cs}O_2$ (mmHg) | 0.58 | 0.17 |
| $P_{cs}CO_2$ (mmHg) | 0.80 | 0.35 |
| $pH_{cs}$ | 0.79 | 0.85 |
| $BE_{cs}$ (mmol/l) | 0.53 | 0.17 |
| $SBIC_{cs}$ (mmol/l) | 0.63 | 0.16 |
| $Lactate_{cs}$ (mmol/l) | 0.06 | 0.03 |
| $Isoflurane_{cs}$ (μg/dl) | | |

COP is calculated by linear extrapolation of the instantaneous diastolic P-F relationship in the respective grafts to zero flow. Similarly, but without extrapolation to zero flow, CVR can be assessed by the slope of the instantaneous diastolic P-F relationship. The

determination COP in IMAB grafts for calculating CPP and CVR revealed that both CSP and LVEDP (generally used as measures of downstream pressure to calculate CPP and CVR) underestimate the effective downstream pressure and, consequently, CVR.

The concept of using COP to define the effective downstream pressure is convincing because basic physiology predicts that blood flow ceases if the difference between the upstream and downstream pressure in a vascular tree equals zero; thus, the arterial pressure at zero flow represents the effective downstream pressure of organ blood flow (*Hoffman & Spaan, 1990*).

Earlier experimental and clinical studies investigating the effects of massive coronary vasodilatation on diastolic P-F demonstrated that the linearity of the relationship between pressure and flow velocity was not influenced. The slope increased and COP decreased after vasodilatation in these studies (*Dole et al., 1984*; *Klocke et al., 1981*; *Nanto et al., 2001*). Intracoronary injections of angiographic contrast medium or adenosine triphosphate depressed vasomotor activity and induced an atrioventricular blockade followed by a non-physiological increase in diastole.

In contrast to these studies, we found a remarkable decrease in CVR only when COP was used to calculate the effective perfusion pressure and CVR. As expected, the calculation of CVR using the instantaneous diastolic P-F relationship yielded identical results. Using CSP and LVEDP as downstream pressures to calculate CPP and CVR revealed unchanged CPP and CVR. CSP and LVEDP may not adequately reflect downstream pressure (*Kazmaier et al., 2006*). In earlier studies, blood flow velocity measurements were performed (in contrast to our study in which blood flow was assessed). The relative accuracy of flow probes has been described and validated (within ± 2%)(*Beldi et al., 2000*; *Groom et al., 2001*). Flow velocity is linearly related to flow only if the diameter of the vessel in which the measurements are performed is calculated. Studies have demonstrated that the cross-sectional areas of epicardial coronary vessels are nearly independent of pressure and flow (*Douglas & Greenfield, 1970*; *Klocke, Ellis & Orlick, 1980*). Thus, differences in results are probably not due to different measurement techniques but to different study conditions.

Furthermore, it remains questionable whether the CVR and COP results assessed in the previously mentioned studies during maximal coronary vasodilatation and atrioventricular blockade reflect physiologic conditions in human coronary circulation as presented in our study.

Most studies on intraoperative myocardial blood flow measurements are focused on graft patency (*D'Ancona et al., 2000*; *Leung et al., 1991*; *Takami & Ina, 2001*; *Walpoth et al., 1998*). In our first study, we demonstrated the feasibility of calculating COP, CPP and CVR by assessing the instantaneous P-F relationship in coronary bypass grafts (*Kazmaier et al., 2006*). In another study, a negative correlation was observed between COP and residual myocardial viability after angioplasty in patients with acute myocardial infarction (*Shimada et al., 2003*). In a group of similar patients, short- and long-term myocardial outcomes were closely correlated with the deceleration time of diastolic coronary flow velocity (*Furber et al., 2004*; *Yamamuro et al., 2002*). Several studies

have demonstrated the myocardial protective effects of isoflurane in patients undergoing CABG surgery. Nevertheless, in our study, isoflurane did not influence COP (in contrast to CPP and CVR). Additionally, CPP was considerably overestimated if CSP or LVEDP, which were not influenced by isoflurane, were used in the calculations.

In addition to providing a more valid calculation of CPP using COP for downstream pressure, our study also has clinical implications. After short-term application of isoflurane with sub-anesthetic concentrations, COP was unchanged, but MDAP decreased significantly, resulting in a remarkable decrease in CPP. Despite this decrease in CPP, coronary flow remained unchanged, which is due to a considerable decrease in CVR. These effects could only be measured by the calculation of CVR using the diastolic instantaneous P-F relationship (in contrast to the standard formula). Undiminished flow was accompanied by an unaltered CI. Nevertheless, we observed increased $ScsO_2$, which is an indicator of hyper-perfusion.

Our study had a few limitations that may affect the interpretation of these results.

The second measurement in the isoflurane group was performed after 15 minutes of isoflurane application. The expiratory isoflurane concentrations reached 95% of the inspiratory concentrations indicating steady state conditions in the isoflurane blood concentrations at 1.0 $MAC_{sedative}$. Nevertheless, we cannot exclude the possibility that the myocardial uptake of isoflurane was incomplete resulting in unstable concentrations in the myocardial tissue. Studies on long-term applications and dose-response relationships of isoflurane may provide further insights on the effects of isoflurane on the instantaneous P-F relationship in arterial coronary bypass grafts.

## CONCLUSIONS

Short-term application of isoflurane with sub-sedative concentrations markedly increased the slope of the instantaneous diastolic P-F relationship in arterial coronary bypass grafts. This finding implies a distinct decrease in CVR in patients undergoing CABG surgery that cannot be observed with the conventional CVR calculations using MDAP as the upstream pressure and LVEDP or CSP as the downstream pressure. The simultaneous decrease in myocardial oxygen extraction supports the validity of CVR calculation using the slope of the diastolic P-F relationship possibly indicating hyper-perfusion of the myocardium. Thus, the technique provides a more rational approach for the measurement of regional coronary vascular perfusion. Independent of the anesthetic, CSP and LVEDP greatly underestimated the effective downstream pressure and COP leading to a systematic overestimation of CPP. This result can be explained by a vascular waterfall phenomenon in the coronary circulation. However, in contrast to CVR, COP (the more reliable downstream pressure) is not influenced by isoflurane.

## ACKNOWLEDGEMENTS

The authors thank the staff of the Department of Anesthesiology all of whom were involved in patient care and monitoring.

### Funding

The authors received no funding for this work.

### Competing Interests

The authors declare that they have no competing interests.

### Author Contributions

- José Hinz conceived and designed the experiments, performed the experiments, analyzed the data, wrote the paper, prepared figures and/or tables, reviewed drafts of the paper.
- Ashham Mansur conceived and designed the experiments, analyzed the data, wrote the paper, prepared figures and/or tables, reviewed drafts of the paper.
- Gerd G. Hanekop conceived and designed the experiments, performed the experiments, contributed reagents/materials/analysis tools, wrote the paper, reviewed drafts of the paper.
- Andreas Weyland conceived and designed the experiments, performed the experiments, contributed reagents/materials/analysis tools, wrote the paper, reviewed drafts of the paper.
- Aron F. Popov conceived and designed the experiments, analyzed the data, wrote the paper, prepared figures and/or tables, reviewed drafts of the paper.
- Jan D. Schmitto conceived and designed the experiments, performed the experiments, wrote the paper, reviewed drafts of the paper.
- Frank F. G. Grüne conceived and designed the experiments, analyzed the data, wrote the paper, reviewed drafts of the paper.
- Martin Bauer conceived and designed the experiments, analyzed the data, wrote the paper, reviewed drafts of the paper.
- Stephan Kazmaier conceived and designed the experiments, performed the experiments, wrote the paper, prepared figures and/or tables, reviewed drafts of the paper.

### Human Ethics

The following information was supplied relating to ethical approvals (i.e., approving body and any reference numbers):

This study was approved by the University of Goettingen ethics committee in Goettingen, Germany (12/4/04).

### Clinical Trial Ethics

The following information was supplied relating to ethical approvals (i.e., approving body and any reference numbers):

This study was approved by the University of Goettingen ethics committee in Goettingen, Germany (12/4/04).

## Data Deposition

All relevant data can be found within the manuscript.

## Clinical Trial Registration

The following information was supplied regarding Clinical Trial registration:

DRKS00008892.

## Supplemental Information

Supplemental information for this article can be found online at http://dx.doi.org/10.7717/peerj.1619#supplemental-information.

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
