# Peer review of "Influence of isoflurane on the diastolic pressure-flow relationship and critical occlusion pressure during arterial CABG surgery: a randomized controlled trial"

_PeerJ, doi:10.7717/peerj.1619_

## Round 0.1 · original submission · Major Revisions

Dear authors:

After reading the paper and the comments of the reviewers (particularly those of the third one), I think there are some major issues which you have to solve. Some of them need to be justified and I do not know if your sample size will be enough for your objective. Please, try to perform all the calculations indicated by the reviewer #3 as well as the rest of the comments of all the reviewers.

Reviewer 1 ·

Basic reporting

No Comments

Experimental design

No Comments

Validity of the findings

No Comments

Additional comments

1. The researchers found a remarkable decrease in CVR only when COP was used to calculate the effective perfusion pressure and CVR. The calculation of CVR using the instantaneous diastolic P-F relationship yielded identical results. However, COP calculated by linear extrapolation of the instantaneous diastolic P-F relationship in the respective grafts to zero flow was not influenced by isoflurane. Adequate interpretation about the unchanged of COP should be given.
2. In the research just twenty patients were studied, so the test sample was small. In addition, long-term applications and dose-response relationships of isoflurane should be studied in further research.
3. Figure which reflect instantaneous diastolic P-F relationship should be given in the article. So the slope of the instantaneous diastolic P-F relationship can be conclude directly.

·

Basic reporting

Excellent study
Unstructured abstract
Please describe COP in detail in introduction,
Please add the Effect of isoflurane on venous graft in discussion if possible
Random allocation is not described in detail (e.g.Type of randomization and sequence)
Sample size calculation is not mentioned

Role of funders is not mentioned

Experimental design

Not mentioned about the the expected complications during study and who will pay.

Validity of the findings

Too many tables with extensive information. Try to limit the number and content of tables.

Additional comments

Excellent topic and a good study on very limited subjects. Need few minor changes

Reviewer 3 ·

Basic reporting

The manuscript was well written. I have some 2 questions:
1. In Calculations section from lines 153-159, could the author provide any references for those methods? Without citation, it's hard to know these calculations were only used in this paper or they were standardized methods.
2. In Table 2b, control vs Isoflurance at T-15 and T-30 were compared, when compared the T-30, did the author control for the values at T-15? To compare values after baseline without controlling for baseline could lead to misleading conclusions.

Experimental design

The design is sound. But a critical flaw in the manuscript is that the authors never justified the chosen sample size of 20, and never reported the power given a sample size of 20. It's very uncommon nowadays to see a clinical trial conducted without these justifications. If the power of sample size of 20 is only 40%, then trial has a 60% chance of type II error, now the question is: should the readers believe more of your 40% chance to get the right findings or should the reader believe more of your 60% chance of not getting the right findings? Given a lower power trial, you may miss a large chance to discover many differences between groups, which will undermine your overall trial findings.

I checked the protocol to see if there is sample size justification or power evaluation, but it's not in English.

Validity of the findings

The findings in this manuscript were at least questionable for the following reasons:

1. Power of the trial is unclear. Lower power trials may fail to discover significant differences between groups, meaning all the author claimed non-significant differences could be significant had the power been larger.
2. The authors conducted dozens of statistical tests but never mentioned about controlling type I error, meaning their claimed significant differences between groups could be false.
3. The authors conducted only univariate tests and did not control for any covariates in all the tests without any justification. Testing without controlling for any convariates may lead to discovery of false difference.

Overall, given the unjustified small sample size used in the trial and the questionable statistical tests conduced, it's challenging to believe the findings of this article.

Additional comments

No Comments

---

## Round 0.2 · accepted · Accept

Dear authors,

I consider your paper has high standards to be published in PeerJ.

Congratulations!

With respect and warm regards,
Dr Palazón-Bru (academic editor for PeerJ)

Reviewer 1 ·

Basic reporting

No Comments

Experimental design

No Comments

Validity of the findings

No Comments

·

Basic reporting

No comments

Experimental design

No comments

Validity of the findings

No further comments

Additional comments

Excellent study. Suggested changes done where required. I would accept this article for publication